# Phytochemical Differentiation of Saffron (*Crocus sativus* L.) by High Resolution Mass Spectrometry Metabolomic Studies

**DOI:** 10.3390/molecules26082180

**Published:** 2021-04-10

**Authors:** Evangelos Gikas, Nikolaos Stavros Koulakiotis, Anthony Tsarbopoulos

**Affiliations:** 1Department of Chemistry, National and Kapodistrian University of Athens, 15771 Athens, Greece; vgikas@chem.uoa.gr; 2GAIA Research Center, Bioanalytical Department, The Goulandris Natural History Museum, 14562 Kifissia, Greece; koulak13@gmail.com; 3Department of Pharmacology, Medical School, National and Kapodistrian University of Athens, 11527 Athens, Greece

**Keywords:** *Crocus sativus* L. (Iridaceae), saffron, reversed-phase UPLC, metabolomics, HR MS

## Abstract

The metabolite profiling of saffron (*Crocus sativus* L.) from several countries was measured by using ultra-performance liquid chromatography combined with high resolution mass spectrometry (UPLC-HR MS). Multivariate statistical analysis was employed to distinguish among the several samples of *C. sativus* L. from Greece, Italy, Morocco, Iran, India, Afghanistan and Kashmir. The results of this study showed that the phytochemical content in the samples of *C. sativus* L. were obviously diverse in the different countries of origin. The metabolomics approach was deemed to be the most suitable in order to evaluate the enormous array of putative metabolites among the saffron samples studied, and was able to provide a comparative phytochemical screening of these samples. Several markers have been identified that aided the differentiation of a group from its counterparts. This can be important for the selection of the appropriate saffron sample, in view of its health-promoting effect which occurs through the modulation of various biological and physiological processes.

## 1. Introduction

*Crocus sativus* L. is a species of flowering plant of the Crocus genus which grows in the Mediterranean, east Asia and the Irano-Turanian region. *Crocus sativus* L. is a member of the Iridaceae family, and is cultivated worldwide due to the use of its dried styles (the uppermost colored part of which is referred to as stigma), not only as a spice (saffron), but also in health management since ancient times [1,2]. Saffron is a perennial spicy herb which is difficult to be cultivated since it demands a special climate and soil conditions. The earliest apparent reference to *Crocus sativus* L. cultivation goes back to around 2300 BC, and a saffron harvest is shown in a Minoan fresco painting in the Knossos palace of Minoan Crete dated from 1600–1500 BC [3]. It is also seen in a fresco in Akrotiri on the Greek island of Thera dated back in 1627 BC [4], which depicts the flowers being picked by young girls and monkeys. The most plausible ancestor of *Crocus sativus* L. was *Crocus cartwrightianus* [5,6,7], as derived from morphological [5], cytological [8] and molecular analyses [9].

Saffron is the spice derived from the flower of *Crocus sativus* which is comprised of the three red stigmas included in the flower that are consequently collected and dried under special conditions to produce the final saffron as a spice. Crocus contains more than 150 volatile aromatic substances that afford its distinctive aroma, and a large number of non-volatiles such as carotenoids including zeaxanthin, lycopene, as well as various α- and β-carotenes, glycosides, monoterpenes, aldehydes, flavonoids, anthocyanins, vitamins (especially riboflavin and thiamine) and amino acids [10]. The four main bioactive constitutes of saffron stigma are crocetin, crocins, picrocrocin and safranal [11]. In previous studies, the volatile compounds of saffron samples have been characterized by gas chromatography–mass spectrometry (GC–MS) methods [12,13], and have been evaluated as markers of geographic differentiation [14]. In a recent study, the metabolite profiling of three different parts of *Crocus sativus* L., i.e., tepals, stigmas and stamens, was measured by ultra-performance liquid chromatography (UPLC) coupled to hybrid quadrupole time-of-flight mass spectrometry (QqTOF MS), which provided the diverse chemical characteristics of the parts of the flower [15]. In the last 20 years, there has been an increasing amount of scientific data on saffron extract or its constituent’s biological activity and health-promoting properties, including use as an anticonvulsant [16], anti-inflammatory [17], anti-tumor [18], anti-oxidant [19], antiatherogenic [20] and has shown antidepressant [21] activity, as well as enhancement of learning and memory capacity [22,23]. Therefore, saffron and its ingredients can be useful in the treatment of a variety of diseases such as neurodegenerative disorders, blood pressure abnormalities, acute and/or chronic inflammatory disease and coronary artery disease. The main bioactive constituents in saffron are the crocins existing mainly in the stigmas, and they are mono- and bis-esters of crocetin with glucose, gentiobiose and/or gentiotriose [24]. The esterification of crocetin with varying number of hydrophilic gentiobiose(s), or any other sugar precursors, renders the carotenoid derivative water soluble, a property that gives crocus its pigmentation properties. The complexity of crocins may result from the incorporation of various sugars connected to crocetin such as glucose, gentiobiose, gentiotriose and neapolitanose, the number carbohydrate units (1–5), the number of glycosylation sites (1,), their linkage to the acid moiety of the carotenoid, or even by the varying number of repeating units in crocetin [25]. Therefore, there is a huge range of crocins, with most of them being found only in trace amounts. To date, the majority of studies performed support the therapeutic potential of crocins in treating aging and age-related neurodegenerative disorders. The major crocin component, *trans*-crocin-4 (TC4; bis-gentiobiosyl-E-crocetin) possesses numerous pharmacological activities including antihypertensive [26], anxiolytic [27] and neuroprotective [28] activity. In particular, TC4 has shown the highest inhibitory potential towards reducing or even preventing amyloid oligomerization [29], which is considered as one of the main causes of Alzheimer disease (AD) progression [30]. Conversely, the characteristic flavor of saffron is due to picrocrocin, the glucoside of the terpene aldehyde safranal, which comprises more than 4% of the dry weight of saffron. Safranal is produced by the oxidative cleavage of the carotenoid zeaxanthine, and accounts for more than 70% of the volatile fraction of the spice. Saffron is likely the most expensive spice, owing to its very limited geographical spread and the difficulty of its collection. The higher content of the analyzed saffron samples in crocin, picrocrocin and safranal indicates the higher value of saffron.

We know that plant secondary metabolites are a group of naturally occurring compounds biosynthesized by differing biochemical pathways, and their plant content and regulation is strongly amenable to environmental influences as well as to potential herbal predators [31,32,33]. Moreover, the level and type of secondary metabolites is strongly influenced by the geoclimatic characteristics of the cultivation area as well as the preparation procedures and traditions followed in that area. Therefore, there is a necessity to identify the content of bioactive compounds in collected saffron samples and explore whether there is any correlation between different geographical regions and the contents of the bioactive compounds.

The aim of the current study is to explore the chemical space of *Crocus sativus* L. from different geographical regions in order to spot chemotaxonomic differences in the indigenous species. This could potentially aid towards our understanding of the plant’s biochemistry but also the precise evaluation of its cultivation in diverse environments. We employ a metabolomics methodology based on UPLC high resolution mass spectrometry (HR–MS) in order to provide detailed information on the metabolite profiling of several samples of *C. sativus* L. from Greece and six other countries/areas: Italy, Morocco, Iran, India, Afghanistan and Kashmir. The metabolomics approach was deemed to be the most suitable choice in order to evaluate the enormous array of putative metabolites among the saffron samples studied, and thus provide a comparative phytochemical screening of the saffron samples studied.

## 2. Results and Discussion

### 2.1. UPLC-MS Analysis

A representative base peak LC–MS chromatogram of the stigma extract is shown in Figure 1. Several peaks have been annotated in the early eluting part of the chromatogram with the names of the putative metabolites shown in Table 1.

The mapping of the chemical potential of saffron according to its geographical spread is of significant importance in the effort to understand evolutional pressure exerted on the species, as well as to provide a chemotaxonomic tool towards the distribution of variants around the world (Figure 2). The content of the active secondary metabolites has an apparent effect on the quality as well as the medical properties of saffron. Furthermore, the extreme cost of the *Crocus sativus* L. stigmas, considered as the most valuable spice, mandates their accurate fingerprinting in order to control its quality. In order to capture the chemical space involved, as well as to compare the species in a holistic manner, an HRMS metabolomics approach was taken.

Specimens from seven representative regions capturing saffron’s biodiversity around the world, namely Iran, Greece, Italy, Afghanistan, Kashmir, Morocco, and India were analyzed. The pairwise comparison of all possible combinations of the saffron specimen was employed, as this approach effectively highlights the differences between species regardless of their magnitude. In the case of a “total” comparison, the model would be dominated by the largest variance, and therefore would be biased towards the species with the largest pairwise differences. Figure 2 depicts the regions of collected specimens of saffron samples. The associated clusters are illustrated and interconnected in different colors, such as those between Greece and Morocco with saffron samples in orange, whereas those between India and Afghan specimens are indicated by green. These interconnections have been revealed in the multivariate statistical analysis (Figure 3).

### 2.2. Multivariate Statistical Analysis

In order to gain insight into the chemical space covered by the genus, as well as to discover trends and spot any possible outliers, a Principal Component Analysis (PCA) analysis was performed employing Pareto scaling. No significant clustering was apparent, but also no outlier values were detected. The R^2^ was 0.707 with the Q^2^ being 0.49 (7-fold cross validation). In order to further explore potentially significant markers among the samples, orthogonal Projections to Latent Structures Discriminant Analysis (oPLS-DA) was applied to enhance separation among the groups in PCA. The oPLS-DA algorithm was used in order to explore for underlying associations existing in the data, as it is considered a more efficient discriminating algorithm. Using Par scaling, clear clustering has been observed showing five clusters of samples, as depicted in Figure 3.

Thus, each one of the species found in Italy Iran and Kashmir were allocated to unique clusters as their only members, while Greece formed a cluster with Morocco, and Indian and Afghan saffron species formed another cluster. The model exhibited excellent fitting (R^2^ = 0.896) and predictive power (Q^2^ = 0.688).

In order to focus on differentiating metabolites between species, all pairwise oPLS-DA models between the five groups were constructed. Ten models were constructed as shown in Table 1. All models were validated by permutation testing, whereas ANalysis Of VAriance testing of Cross-Validated predictive residuals (CV-ANOVA) was used to verify the statistical significance of the model (*p* < 0.05). To verify the validity of the multivariate analysis concerning the generated models, the Hottelings T2 and the DModX were evaluated, and were considered as valid when no value exceeds the d-critical level set to 0.05. The residuals normality was also considered and examined for values deviating from normality. In order to discover the most influential features for the construction of each model, the corresponding S-plot was evaluated in every case, where the most differentiated metabolites for each compared group can be distinguished. Therefore, the Paretto based models were considered, whereas the VIP values were also considered in the process. As a rule of thumb, the five major features from each group of each pairwise comparison were investigated. All features identified in the manuscript are at identification level four [34]. The big three MS methodologies employed (MS, MS/MS and HR MS) gave access to corresponding fragmentation used for the assignment of probable structure through MS/MS. Features were attributed to metabolites based on multiple criteria, i.e., that the accurate mass should not deviate more than 10 ppm from its theoretical value, the isotopic pattern should show a score of >90, while the MS/MS fragments should be present with unit resolution (as they were acquired in the linear ion trap using a parallel scan). The second column in Table 1 lists the accurate protonated molecular ions MH^+^ with their respective retention times, whereas the third column lists the putative metabolites and the corresponding fragment ions (in brackets) obtained in the linear ion trap using a parallel scan. The metabolites tabulated in Table 1 are the ones that are upregulated in the first pair (e.g., Greece and Morocco) and downregulated in the counterpart pair of the comparison (e.g., India and Afghan). It should be noted that there was no need to employ crocin standards available in our laboratory for the identification of putative metabolites, because none of them was identified to be significantly differentiated between the saffron samples analyzed, thus not assisting the discrimination of their geographical origin.

### 2.3. Geographical Region Differentiation

The results show that saffron samples were differentiated according to the geographical region of their collection. Interestingly, crocus from distant areas e.g., Greece and Morocco, exhibit more pronounced similarities compared to neighboring regions such as Greece and Italy. This could be attributed to the pivotal impact of microclimatic conditions rather than considering the wider geographical area of cultivation. The Moroccan climate is typically Mediterranean, resembling the Greek weather, even in the Atlantic coast of the country. Nevertheless, the proximity of Greece to Italy and the fact that they are both Mediterranean countries should indicate a large degree of similarity for the species. The Italian saffron shows the same degree of differentiation to the Greek, Moroccan and Iranian species. This reflects that the geoclimatic characteristics, along with the different preparation practices followed in the cultivation area determine the chemical composition of the final product [14,31]. The Greek saffron is collected in a very narrow area (a village in Macedonia province called Krokos from the name of the plant) where the conditions are likely to be the same when compared to the conditions in the collection area of Morocco. Indeed, this is also reflected to the Asian derived species. Thus, the Indian and Afghan saffron are more similar to the Greek and Moroccan species in terms of chemical profile when compared to either Kashmir region or their Iranian counterparts. It should be noted that the Iranian and Kashmir species are more differentiated in terms of chemical components content, despite their proximity.

Another issue that should be noted is the genetic profile of the species. Considering that they are cultivated plants rather than native, it seems that the phylogenetic association is more closely related to the human intervention than to their historically driven distribution. Thus, an assumption that needs further verification is that the Iranian or the Kashmir branch were transferred by merchants to the countries that were in financial contact. The Iranian/Middle East axis has a strong impact on the human financial relations, and it seems that the same holds true for the Kashmir/India branch. The *Crocus sativus* L. species were cultivated and integrated in the areas from the Mediterranean basin to Iran/Middle East and India/Kashmir, and both their similarities and differences clearly reflect the international trade and financial relations between countries that were geographically distant.

### 2.4. Saffron Components in the Saffron Samples from Various Regions

Saffron contains more than 150 volatile and aroma-yielding compounds. It also has a number of nonvolatile active components [10], many of which are carotenoids, including zeaxanthin, lycopene, and various α- and β-carotenes. The wealth (plethora) of chemical components in saffron poses complexity for its analysis. Several chromatographic and mass spectrometric methods have been developed for the quantitation of the main bioactive ingredients of saffron, such as crocins and picrocrocin [14,35,36,37,38]. The content of the active secondary metabolites has an effect on the quality and efficacy of saffron [14,26]. In view of the limitations of other techniques, ultra-performance liquid chromatography (UPLC-HR MS) has been considered to be the most suitable method to analyze the constituents of saffron extracts. In our study, UPLC-HR MS provided the metabolomic profile of the saffron samples affording high sensitivity and retention time reproducibility. The UPLC-HR MS and multivariate statistical analysis were combined to analyze saffron stigmas. Our results showed that chemical characteristics of saffron were apparently diverse, which mainly arose from the different geoclimatic characteristics inherent to the territory of cultivation. Moreover, changes in the preparation procedures, i.e., flower collection, separation/drying and conservation of stigmas, may strongly modify the final composition of chemical components present in the stigma.

### 2.5. Marker Discriminating Power

In order to discover generalized markers of discriminating the saffron species, the cross-tab (Table 2) has been created. The intention was to identify a single feature or even a couple that could differentiate a group from its counterparts. There were two clusters, those of Indian-Afghan and those in the Greek-Morocco saffron that were considered as belonging to the same group. Thus, no generalized marker was found for discriminating the Indian-Afghan saffron from the other groups, however the 252.1061_0.59 and 472.1734_3.24 ions could differentiate the four of the five groups. The former mass signal corresponds to the (M+NH_4_)^+^ adduct ion of 3,4-Epoxybisabola-7(14),10-dien-2-one (EDO; C_15_H_22_O_2_) marker, whereas the latter corresponds to the (M+Na)^+^ adduct ion of astragalin (C_21_H_20_O_11_). Conversely, the putative metabolite of tomentogenin [12] with MH^+^ signal 369.15_2.95 could be used to discriminate three of the five groups in the case of the Iranian stigmas. In the case of the Greek-Morocco saffron, the results were even less general, and no molecular species was found to show such a capacity. The Italian varieties could be clearly separated from the others using the 169.1211_3.66 and the 353.1548_3.66 ions corresponding to the MH^+^ of vanillic acid and the (M+Na)^+^ adduct ion of picrocrocin, respectively. Finally, the 611.157_4.14 ion corresponding to the MH^+^ ion of kaempferol-di-glucoside was found it can be employed to distinguish the Kashmir saffron. These results indicate that a combination of markers should be employed which necessitates the use of hyphenated separation methodologies (e.g., LC–MS) for achieving the screening of *saffron* extracts, as well as probing the *saffron* for the presence of adulterants [26].

## 3. Materials and Methods

### 3.1. Chemical Reagents and Standards

Methanol (MeOH) and acetonitrile (ACN) of HPLC grade were supplied from were supplied from Carlo Erba (Milano, Italy) and Fisher Scientific (Pittsburgh, PA, USA), respectively. Trifluoroacetic acid (TFA) was obtained from Αcros Organics (Fair Lawn, NJ, USA) and water was purified using Μilli-Q (RG) filter systems from Millipore Corporation (Billerica, MA, USA).

### 3.2. Sample Collection

*Crocus Sativus* L. dried styles (saffron) were kindly supplied by the Cooperative De Safran, (Krokos Kozanis, West Macedonia, Greece). The *saffron* samples from Morocco, Italy, Iran, India, Kashmir and Afghanistan were purchased from Sahar Saffron Company (Cleveland, OH, USA). All samples were kept refrigerated at 5 °C until analysis.

### 3.3. Sample Preparation

50 mg of *saffron* stigmas were soaked in methanol water 1:1 (*v*/*v*) for 200 days in dark under ambient temperature. The stigmas were extracted with 10mL MeOH:water (1:1, *v*/*v*) for 24 h at 25 °C in the absence of light with continuous stirring, and then centrifuged, filtered through a 0.2-μm filter and evaporated to dryness employing a Speed Vac system (Labconco Corp., Kansas City, MO, USA). Samples were reconstituted in MeOH:water (1:1, *v*/*v*), transferred to 1.5 mL autosampler vials and an appropriate volume was injected to the LC–MS system.

### 3.4. UPLC—HR MS Metabolomics Analysis

A quality control sample (QC) taken from all samples was prepared in order to periodically assess the reproducibility of the measurements. The separation of the analytes contained in the saffron samples was achieved with a Fortis UPLC C_18_ column (2.1 mm × 100 mm, 1.7 µm, Fortis Technologies Ltd., Cheshire, UK). The hyphenated LC-HRMS system comprised of an Accela UHPLC equipped with an autosampler, a vacuum degasser, a binary pump and a temperature-controlled column (Thermo Scientific, Germany) coupled to an Orbitrap Discovery XL, which was equipped with an IonMAX ion source (Thermo Scientific, Bremen, Germany). The mobile phase consisted of 0.1% aq. formic acid (*v/v*) (solvent A) and 0.1% formic acid in LC–MS grade ACN (*v/v*) (solvent B). The gradient program was for solvent B: 5% at 0 min, 5% at 3 min, 95% at 24 min, 95% at 26 min, 5% at 28 min, 5% at 30 min. The overall analysis time spanned for 30 min, whereas the injection volume was 5 µl keeping a flow rate of 400 µl min^−1^. The positive ionization ESI mode was used using a mass range of 100–1000 amu. The “big three” approach, employing parallel scans, was used. The samples were centrifuged using a Mikro 200R centrifuge (Hettich Lab Technology, Tuttlingen, Germany), and for the solvent evaporation was performed on a GeneVac HT-4X EZ-2 series evaporator Lyospeed ENABLED (Genevac Ltd., Ipswich, UK).

### 3.5. Statistical Analysis

The raw data were imported to the Mzmine 2.51 [39] and the Automated Data Analysis Pipeline (ADAP) pipeline was employed, using the wavelets methodology for the chromatogram deconvolution as implemented to ADAP [40]. The feature list was analyzed by SIMCA 14.1 (Umetrics, Umea, Sweden) for the construction of the multivariate models, whereas all univariate analyses were performed by Jamovi. The multiple correction for t-testing, which used the false discovery rate approach, was performed by the qvalue R package [41]. All multivariate models were validated by n-fold as well as by permutation testing, employing 100 random permutations. The CV-ANOVA was used with *p* < 0.05 to verify the validity of the produced models [42].

### 3.6. Feature Identification

The peak list MS features were annotated using the KEGG, CheBI, MetaCyc, LIPIDMAPS, FOR-IDENT, and HMDB libraries. The Met-Frag online version was employed [43] for the annotation and an additional home-assembled MS library was also used.

## 4. Conclusions

The metabolomic profiling of saffron (*Crocus sativus* L. stigma) from different geographical regions, employing UPLC-HR MS analysis combined with multivariate statistical analysis, provided evidence that the phytochemical content in the samples of *C. sativus* L. was diverse in the different countries of origin. This diversity apparently arises from the different geoclimatic characteristics of the area of cultivation in combination with the distinct preparation procedures in the respective countries. Our results indicated that there are characteristic ions that could differentiate a certain group from its counterparts such as the Indian-Afghan and the Greek-Morocco saffron samples that could be considered to belong to the same group. The metabolomics approach was deemed to be the most suitable choice in order to evaluate the enormous array of putative metabolites of saffron, and thus provide a comparative phytochemical screening among the saffron samples studied. In addition, this UPLC-HR MS-based metabolomics approach could be also employed for probing possible adulteration of saffron samples. In view of saffron’s health-promoting effects through the modulation of various biological and physiological processes, the selection of the appropriate saffron sample guided by the respective metabolomic profiling could be an important step. That, in turn, may aid the emerging popularity and interest in alternative medicine-based treatments within health practices.

## Figures and Tables

**Figure 1 molecules-26-02180-f001:**
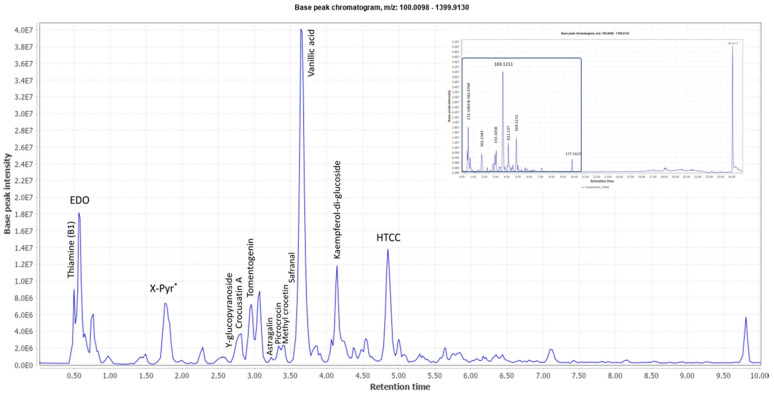
Characteristic ultra-performance liquid chromatography coupled to high resolution mass spectrometry (UPLC-HR MS) base peak chromatogram of the stigma extract from the *C. sativus* L. sample from India. This represents the early eluting part of the chromatogram shown in the inset. Several peaks have been annotated in the early eluting part of the chromatogram with the names of the putative metabolites shown in Table 1. * X-Pyr: 3,4,5-Trimethoxyphenol-1-O-[β-d-apiofuranosyl-(1→6)]]-β-d-glucopyranoside.

**Figure 2 molecules-26-02180-f002:**
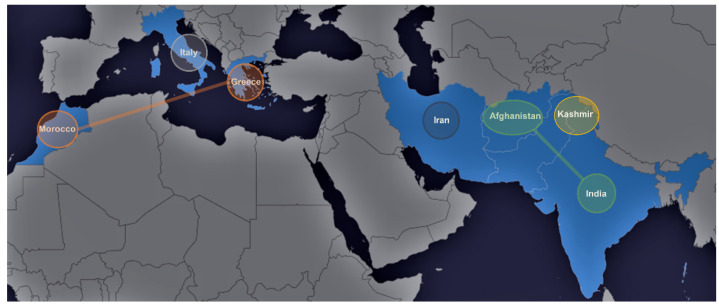
Regions of collected specimens of saffron samples along with their associated cluster.

**Figure 3 molecules-26-02180-f003:**
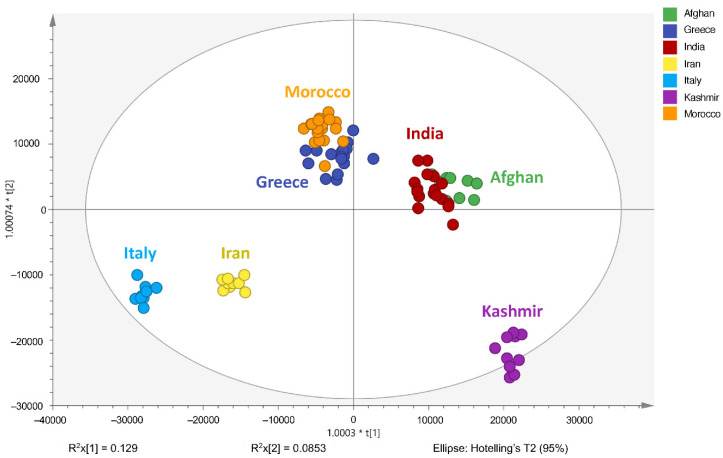
Orthogonal Projections to Latent Structures Discriminant Analysis (oPLS-DA) (**Par scaling**) of the seven saffron species examined.

**Table 1 molecules-26-02180-t001:** Cumulative table of the **A.** Multivariate statistics **B.** Identifying discriminating features and **C.** Putative metabolite annotation with the calculated protonated molecular ion (MH^+^) or the adduct ion mass signals shown in parenthesis, whereas the fragment ions are denoted in curly brackets.

A. Scores Plot and S-Plot	B. Features (Accurate MH^+^_Retention Time)	C. Putative Metabolite (Calculated Mass) * {Fragment} ^1^
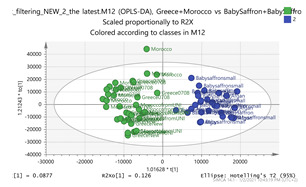 Greece + Morocco vs. India + Afghan Par 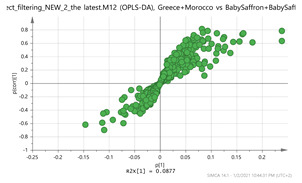	307.1003_4.95 365.1196_3.36 369.15_2.95 383.1298_2.85 472.1734_3.24 vs. 252.1061_0.59 367.135_3.42 369.1508_1.49 305.0824_0.71 167.1057_1.49	EGC: C_15_H_14_O_7_ (307.0818) {263, 139} Methyl crocetin: C_21_H_26_O_4_ (MNa^+^ = 365.1729) {118} Tomentogenin: C_21_H_36_O_5_ (369.2641) {232} Astragalin: C_21_H_20_O_11_ (MNa^+^ = 472.0982) {287, 145} EDO: C_15_H_22_O_2_ (NH_4_^+^: 252.1963) {140} Bornyl ferulate: C_20_H_26_O_4_ (MK^+^ = 369.1470) {266, 239}
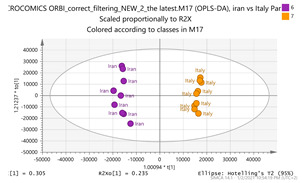 Iran vs. Italy par 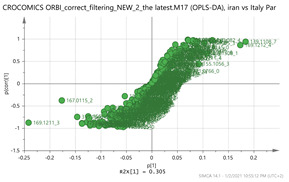	139.1108_7.13 169.1212_4.39 369.1508_1.49203.082_4.75 171.1005_3.58 vs. 169.1211_3.66 177.1629_9.84 353.1548_3.66 167.0115_24.08 265.1216_0.58	HTCC isomer: C_10_H_16_O_2_ (169.1228) Bornyl ferulate: C_20_H_26_O_4_ (MK^+^ = 369.1470) {266, 239} Vanillic acid: C_8_H_8_O_4_ (169.0601) {151, 105} Picrocrocin: C_21_H_26_O_4_ (MNa^+^ = 353.1577) {185, 151} Thiamine (B1): C_12_H_17_N_4_OS (265.1123)^2^ {144, 122}
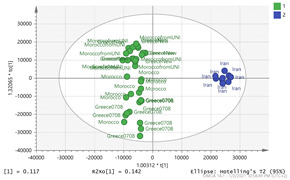 Greece + Morocco vs. Iran Par 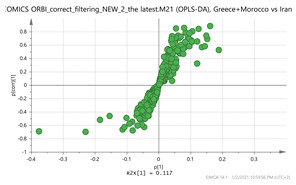	155.1057_2.77 169.1212_4.39 294.1526_0.79 317.1579_2.81 265.1216_0.58 vs. 151.1105_3.67 169.1211_3.66 169.1211_4.86 369.15_2.95 472.1734_3.24	Crocusatin A: C_9_H_14_O_2_ (155.1073) {159} HTCC isomer: C_10_H_16_O_2_ (MH^+^ 169.1228) Y-glucopyranoside: C_15_H_24_O_7_ (317.1600) {383, 261, 196} Thiamine (B1): C_12_H_17_N_4_OS (265.1123)^2^ {144, 122} Safranal: C_10_H_14_O (151.1123) {123} Vanillic acid: C_8_H_8_O_4_ (169.0601) {151, 105} Hydroxy-β-cyclocitral (HTCC): C_10_H_16_O_2_ {169.1228} Tomentogenin: C_21_H_36_O_5_ (369.2641) {232} Astragalin: C_21_H_20_O_11_ (MNa^+^ = 472.0982) {287, 145}
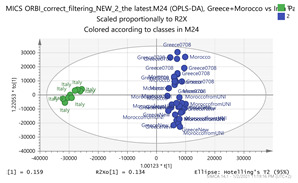 Greece + Morocco vs. Italy Par 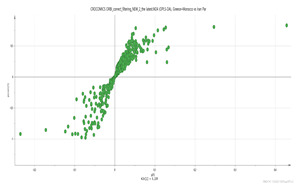	123.1156_3.66 151.1105_3.67 169.1211_4.86 348.1994_3.66 369.15_2.95 vs. 139.1108_7.13 169.1212_4.39 185.1159_3.69 203.082_4.75 171.1005_3.58	Safranal: C_10_H_14_O (151,1123) {123} Hydroxy-β-cyclocitral (HTCC): C_10_H_16_O_2_ (169.1228) Tomentogenin: C_21_H_36_O_5_ (369.2641) {232} HTCC isomer: C_10_H_16_O_2_ (169.1228)
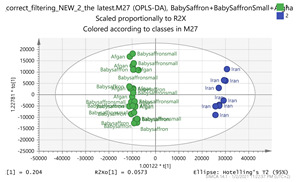 India + Afghan vs. Iran Par 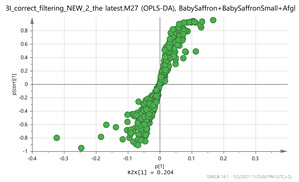	252.1061_0.59 294.1526_0.79 280.1373_0.73 265.1216_0.58 606.229_4.15 vs. 151.1105_3.67 169.1211_3.66 169.1211_4.86369.15_2.95 472.1734_3.24	EDO: C_15_H_22_O_2_ (NH_4_^+^: 252.1963) {140} Thiamine (B1): C_12_H_17_N_4_OS (265.1123)^2^ {144, 122} Safranal: C_10_H_14_O (151.1123) Vanillic acid: C_8_H_8_O_4_ (169.0601) {151, 105} Hydroxy-β-cyclocitral (HTCC): C_10_H_16_O_2_ (169.1228) Tomentogenin: C_21_H_36_O_5_ (369.2641) {232} Astragalin: C_21_H_20_O_11_ (MNa^+^ = 472.0982) {287, 145}
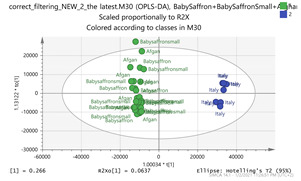 India + Afghan vs. Italy Par 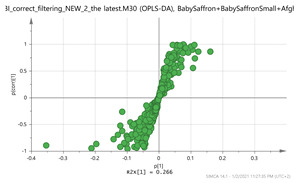	139.1108_7.13 169.1212_4.39 252.1061_0.59 369.1508_1.49 203.082_4.75 vs. 151.1105_3.67 169.1211_3.66 353.1548_3.66 365.1196_3.36 369.15_2.95	HTCC isomer: C_10_H_16_O_2_ (169.1228) EDO: C_15_H_22_O_2_ (MNH_4_^+^ = 252.1963) {140} Bornyl ferulate: C_20_H_26_O_4_ (MK^+^ = 369.1470) {266, 239} Safranal: C_10_H_14_O (151.1123) Vanillic acid: C_8_H_8_O_4_ (169.0601) {151, 105} Picrocrocin: C_21_H_26_O_4_ (MNa^+^ = 353.1577) {185, 151} Methyl crocetin: C_21_H_26_O_4_ (MNa^+^ = 365.1729) {118} Tomentogenin: C_21_H_36_O_5_ (369.2641) {232}
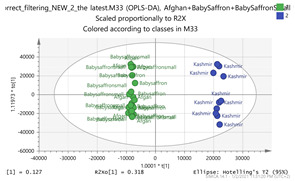 Afghan + India vs. Kashmir Par 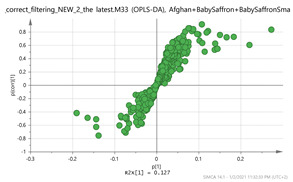	155.1057_2.77 307.1003_4.95 317.1579_2.81 331.173_3.66 364.1944_2.96 vs. 151.1105_3.67 151.1105_4.86 169.1211_3.66 169.1211_4.86 611.157_4.14	Crocusatin A: C_9_H_14_O_2_ (155.1073) EGC: C_15_H_14_O_7_ (307.0818) {263, 139} Y-glucopyranoside: C_15_H_24_O_7_ (317.1600) Picrocrocin: C_16_H_26_O_7_ (331.1757) {185, 151} Methyl crocetin-C_21_H_26_O_4_ (MNa^+^ = 365.1729) {118} Safranal: C_10_H_14_O (151.1123) {123} Vanillic acid: C_8_H_8_O_4_ (169.0601) {151, 105} Hydroxy-β-cyclocitral (HTCC): C_10_H_16_O_2_ (169.1228) Kaempferol-di-glucoside: C_27_H_30_O_16_ (611.1611) {287}
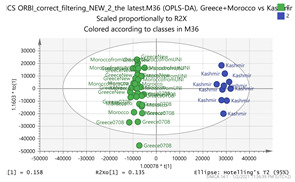 Greece + Morocco vs. Kashmir Par 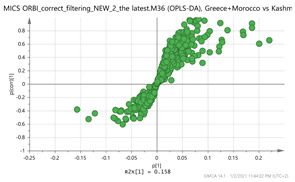	155.1057_2.77 183.1005_3.33 307.1003_4.95 317.1579_2.81 331.173_3.66 vs. 151.1105_3.67 151.1105_4.86 169.1211_3.66 169.1211_4.86 611.157_4.14	Crocusatin A: C_9_H_14_O_2_ (155.1073) EGC: C_15_H_14_O_7_ (307.0818) {263, 139} Y-glucopyranoside: C_15_H_24_O_7_ (317.1600) Picrocrocin: C_16_H_26_O_7_ (331.1757) {185, 151} Safranal: C_10_H_14_O (151.1123) {123} Vanillic acid: C_8_H_8_O_4_ (169.0601) {151, 105} Hydroxy-β-cyclocitral (HTCC): C_10_H_16_O_2_ (169.1228) Kaempferol-di-glucoside: C_27_H_30_O_16_ (611.1611) {287}
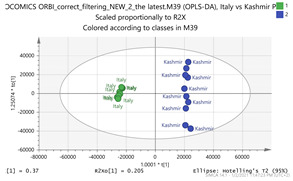 Italy vs. Kashmir Par 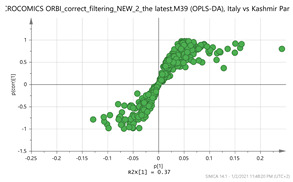	169.1211_3.66 307.1003_4.95 331.173_3.66 353.1548_3.66 369.15_2.95 vs. 169.1212_4.39 252.1061_0.59 369.1508_1.49 203.082_4.75 611.157_4.14	Vanillic acid: C_8_H_8_O_4_ (169.0601) {151, 105} EGC: C_15_H_14_O_7_ (307.0818) {263, 139} Picrocrocin: C_16_H_26_O_7_ (331.1757) {185, 151} Picrocrocin: C_16_H_26_O_7_ (MNa^+^ = 353.1577) {185, 151} Tomentogenin: C_21_H_36_O_5_ (369.2641) {232} HTCC isomer: C_10_H_16_O_2_ (169.1228) EDO: C_15_H_22_O_2_ (MNH_4_^+^ = 252.1963) {140} Bornyl ferulate: C_20_H_26_O_4_ (MK^+^ = 369.1470) {266, 239} Kaempferol-di-glucoside: C_27_H_30_O_16_ (611.1611) {287}
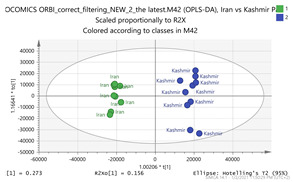 Iran vs. Kashmir Par 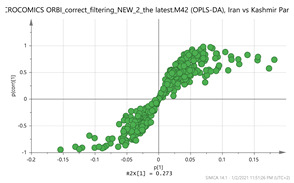	169.1211_3.66 307.1003_4.95 331.173_3.66 369.15_2.95 511.1749_4.94 vs. 252.1061_0.59 294.1526_0.79 280.1373_0.73 265.1216_0.58 611.157_4.14	Vanillic acid: C_8_H_8_O_4_ (169.0601) {151, 105} EGC: C_15_H_14_O_7_ (307. 0818) {263, 139} Picrocrocin: C_16_H_26_O_7_ (331.1757) {185, 151} Tomentogenin: C_21_H_36_O_5_ (369.2641) {232} EDO: C_15_H_22_O_2_ (MNH_4_^+^ = 252.1963) {140} Thiamine (B1): C_12_H_17_N_4_OS (265.1123)^2^ {144, 122} Kaempferol-di-glucoside: C_27_H_30_O_16_ (611.1611) {287}

* The calculated mass values correspond to the protonated molecular ion (MH^+^) ion or the adduct ion (e.g., with Na or K) shown in parenthesis. **^1^** The fragment ions for the selected precursor ions are shown in curly brackets. **^2^** In case of thiamine, the M^+^ ion is observed instead of MH^+^, as the compound has a fixed positive charge. Abbreviations: HTCC: Hydroxy-β-cyclocitral; EGC: Epigallocatechin; EDO: Epoxybisabola-7(14),10-dien-2-one.

**Table 2 molecules-26-02180-t002:** Markers discovered towards the differentiation of the *Crocus sativus* L. stigmas from various destination sources. In bold are marked the common metabolites across at least two destination sources whereas underlined are the ones that could differentiate between all species.

	IND_AFG	GR_MOR	IR	IT	KSH
**IND_AFG**		**307.1003_4.95**	**151.1105_3.67**	151.1105_3.67	**151.1105_3.67**
		365.1196_3.36	**169.1211_3.66**	**169.1211_3.66**	**151.1105_4.86**
		**369.15_2.95**	**169.1211_4.86**	**353.1548_3.66**	**169.1211_3.66**
		383.1298_2.85	**369.15_2.95**	365.1196_3.36	**169.1211_4.86**
		472.1734_3.24	472.1734_3.24	**369.15_2.95**	**611.157_4.14**
**GR_MOR**	**252.1061_0.59**		123.1156_3.66	**169.1211_3.66**	**151.1105_3.67**
	367.135_3.42		**151.1105_3.67**	**177.1629_9.84**	**151.1105_4.86**
	**369.1508_1.49**		**169.1211_3.66**	**353.1548_3.66**	**169.1211_3.66**
	305.0824_0.71		**169.1211_4.86**	**167.0115_24.08**	**169.1211_4.86**
	167.1057_1.49		**369.15_2.95**	**265.1216_0.58**	**611.157_4.14**
**IR**	**252.1061_0.59**	**155.1057_2.77**		**169.1211_3.66**	**252.1061_0.59**
	294.1526_0.79	169.1212_4.39		**177.1629_9.84**	**294.1526_0.79**
	280.1373_0.73	294.1526_0.79		**353.1548_3.66**	**611.157_4.14**
	265.1216_0.58	**317.1579_2.81**		**167.0115_24.08**	280.1373_0.73
	606.229_4.15	265.1216_0.58		**265.1216_0.58**	265.1216_0.58
**IT**	139.1108_7.13	123.1156_3.66	139.1108_7.13		169.1212_4.39
	169.1212_4.39	151.1105_3.67	169.1212_4.39		**252.1061_0.59**
	**252.1061_0.59**	169.1211_4.86	369.1508_1.49		369.1508_1.49
	**369.1508_1.49**	348.1994_3.66	203.082_4.75		**611.157_4.14**
	203.082_4.75	**369.15_2.95**	171.1005_3.58		203.082_4.75
**KSH**	155.1057_2.77	**155.1057_2.77**	**169.1211_3.66**	**169.1211_3.66**	
	307.1003_4.95	183.1005_3.33	307.1003_4.95	307.1003_4.95	
	317.1579_2.81	**307.1003_4.95**	331.173_3.66	331.173_3.66	
	331.173_3.66	**317.1579_2.81**	**369.15_2.95**	**353.1548_3.66**	
	364.1944_2.96	331.173_3.66	511.1749_4.94	**369.15_2.95**	

## Data Availability

The data presented in this study are available in this article.

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
