# Peer review of "Phytochemical Differentiation of Saffron (*Crocus sativus* L.) by High Resolution Mass Spectrometry Metabolomic Studies"

_molecules, 2021, doi:10.3390/molecules26082180_

Round 1

Reviewer 1 Report

COMMENTS AND SUGGESTIONS FOR AUTHORS

In the manuscript entitled “Phytochemical differentiation of Crocus taxa by high resolution mass spectrometry metablolomic studies” by Evangelos Gikas and collaborators, the metabolomic analysis of Crocus sativus from 7 countries was carried out by the authoers using untraperformance liquid chromatography combined with high resolution mass spectrometry (UPLC-HRMS).  The MS data was analyzed using PDA/PLS-DA statistical analysis and algorithm, and Crocus sativus from 7 countries could be separated into 5 groups.  Potential marker ions for the discrimination of Crocus sativus from different group were provided by the authors according to the results of this study as well, which could potentially be applied to check geographical origin of Crocus sativus and prevent food fraud.  This study which accomplished with advance HRMS could provide a great chemotaxonomic research model for foods and herbs and is interesting and worthy of publication in this journal. 

SOME SPECIFIC COMMENTS

  1. The title: The word “Crocus taxa” used here is not appropriate, and it would be better to revised to “Crocus sativus”.  Crocus taxa includes several Crocus species, such as Crocus sativus, Crocus ancyrensis, and Crocus sieheanus.  The differentiation of Crocus taxa refers to the differentiation of different species of Crocus plant.  However, in this study, the material is only focused on one species, Crocus sativus, which indicated that it is unsuitable to use the word “Crocus taxa” to pronoun Crocus sativus.

  2. Page 2, line 76-78: The aim of this study is to differentiate saffron from different region.  However, there were only few sentence describing the reason to conduct this study.  It would be nice to explain the detail regarding the importance to discriminate saffron from geographical different regions.

  3. Page 3, figure 1: The the words in the figure was too blurred to see.  It would be better to improve it.

  4. Page 3, 2.1 UPLC-MS analysis: Please provide all the features and putative metabolite compounds identified by HRMS.  The information of UPLC-MS analysis provided by base peak LC-MS chromatogram was too limited.

  5. Page 3, figure 2: Please explain the meaning of circles in different colors and the linkage between some regions showed in this figure in the figure legend.  The description about figure 2 is not clear.

  6. Page 4, figure 3 and 4: Same as figure 1, the words in the figure is too blurred to see.  It would be better to improve it.

    Figure 4.
    The author depicts the permutation testing of GR-MOR vs IR specifically. What’s the importance of the permutation testing while Figure 3 has already given the main information?
  7. Line 172~173.
    The statement “the chemical profile correlates strictly to the climate conditions of the cultivation area” told firmly while there’s no any direct evidence.  Please provide some references to prove the assumption.

    Line189-191.

    Same question with the former one.  There should be some references to prove the statement “their similarities as well differences definitely reflect the international trade and financial relations between countries.”

Reviewer 2 Report

The present manuscript contains some valuable scientific data, but must be deeply and completely revised to be acceptable for publication on Molecules.

First of all, the authors have to put better in evidence the differences of their LC-MS method with those present in literature and discuss if the used method is able to correctly evaluate all of the most important Crocus metabolites and quality markers. The authors told that saffron contains more than 150 volatile aroma-yielding compounds: how many of them are detectable by UPLC-HR MS? Please add GC references too. The relative abundance of safranal, crocetins, crocins etc. was widely published. Were the present measurements consistent with available data?  Some additional chromatograms should be shown, with the indication of the known molecules detectable in Crocus samples. Retention time variation, stability and alignment must be discussed. 

There's no detail in the paper on identification levels of putative metabolites (see for example dx.doi.org/10.1021/es5002105). Moreover, the putative metabolites listed in table 1 were identified using standards? If yes, how about quantitative determination? If not, how should the second and third columns be read? (overexpressed Vs. underexpressed? - present Vs. absent?)

The sample collection discussion is poor. Specimens description (lines 106-110) has to be moved to Experimental section and geographical description should be more accurate. 

Why no comparison was done between samples from associated clusters? What information do we get from the statistical analysis of not so radically differentiated samples?

Round 2

Reviewer 1 Report

All questions are answered in the revised manuscript.  The compound descriptions listed in Figure 1 panel A and B  are too small.  It is difficult to read as the current pdf file after 200x zoom in.  A higher resolution picture is recommended.

Author Response

We thank the Reviewer for his suggestion. We have now revised Figure 1 with a higher resolution tiff file and larger font size in the compound identification.

Reviewer 2 Report

The revised manuscript molecules-1154440 is far to be a completely new paper. I am surprised that the authors resubmitted the paper in a so short period of time without any substantial modification. I think that the possibility to enhance the quality of the work has not been properly exploited. Anyway the manuscript could be accepted for publication if some important aspects will be implemented. I'm absolutely aware that the aim of the work was an untargeted metabolomics approach, but the conclusions drawn from this study need to be better clarified: what are the practical results obtained? which molecular markers were identified? Why the authors did not clearly indicate which of them are upregulated or downregulated?

My request to compare the analytical results with the literature is not incompatible with the untargeted approach. On the other hand, in order to publish on Molecules, it's strange to exclude a study like this with respect to the broad analytical scenario already published.

Author Response

Reviewer 2

COMMENTS AND SUGGESTIONS FOR AUTHORS

The revised manuscript molecules -1154440 is far to be a completely new paper. I am surprised that the authors resubmitted the paper in a so short period of time without any substantial modification. I think that the possibility to enhance the quality of the work has not been properly exploited.

We thank the reviewer for his willingness for constructive critique. We have revised the manuscript while at the same time we have tried to remain to our main goal, which is to reveal latent connections between the saffron species from the main saffron producing countries around the globe.

Anyway, the manuscript could be accepted for publication if some important aspects will be implemented. I'm absolutely aware that the aim of the work was an untargeted metabolomics approach, but the conclusions drawn from this study need to be better clarified: What are the practical results obtained?

There are more than one practical aspects of the current work. The first is to unveil the putative differentiations in terms of chemical space between selected species from countries that are considered to be the main saffron producing countries. The second aim is to propose markers that could actually differentiate between the studied groups. In our opinion this could be helpful towards the chemotaxonomic investigation of the stigmata, but it could also pave the way towards the establishment of adulteration methodologies. The development of metabolomics-based methodologies and application to plant species thereof, is also of interest as shown in our current study. Thus, a mass spectrometry-based metabolomics methodology established in this work could potentially be used for numerous reasons, some of those described above. This is now indicated in the newly revised manuscript (lines 421-422).

Several markers have been identified that aided the differentiation of a group from its counterparts.

Which molecular markers were identified?

The molecular markers identified are clearly tabulated in Table 1. They resulted from the multivariate statistical analysis of the data obtained by the mass spectrometric analysis and there are identified at the level 4 of identification according to the Schymanski et al. criteria (ref. 34) whereas their identification has been performed by on line databases and the focused experience of our laboratory in the Crocus sativus studies. Identified

Why the authors did not clearly indicate which of them are upregulated or downregulated?

The entries in Table 1 are either the features or metabolites that differentiate the species with the highest degree of statistical confidence (correlation versus covariance). Therefore, the ones that are tabulated are the ones that are upregulated (meaning that they are downregulated in the counterpart of the comparison). The S-plot shown in Table 1provides this particular piece of information, i.e., the upregulated species to each of the two sides of the compared pair; thus, providing the complimentary information indirectly. As an example, please see below the first pair of Table 1showing the upregulated and downregulated pairs. Therefore, we believe that this information would be redundant if it would be indicated in the Table 1. This is now clearly indicated in the newly revised manuscript (lines 201-202 & 214-216).

Scores Plot and S-plot 

Features (accurate Mass_retention time)

Metabolite (Observed MH+ signal) [fragment]

Greece+Morocco vs India +Afghan Par

307.1003_4.95

(upregulated Greece+Morocco)

365.1196_3.36

(upregulated Greece+Morocco)

369.15_2.95

(upregulated Greece+Morocco)

383.1298_2.85

(upregulated Greece+Morocco)

472.1734_3.24

(upregulated Greece+Morocco)

Vs

252.1061_0.59

(upregulated India+Afghan)

367.135_3.42

(upregulated India+Afghan)

369.1508_1.49

(upregulated India+Afghan)

305.0824_0.71

(upregulated India+Afghan)

167.1057_1.49 (upregulated India+Afghan)

EGC - C15H14O7 (307,0818)

Methyl crocetin : C21H26O4 (Na+:365,1729)[118]

Tomentogenin : C21H36O5 (369,2641)[232]

2-cis,6-trans-Farnesyl diphosphate (383.131)

Astragalin : C21H20O11 (MNa+ = 472,0982)[ 177, 145]

EDO : C15H22O2 (NH4+ : 252,1963)

Tomentogenin : C21H36O5 (369,2641)

My request to compare the analytical results with the literature is not incompatible with the untargeted approach. On the other hand, in order to publish on Molecules, it's strange to exclude a study like this with respect to the broad analytical scenario already published.

We agree with the Reviewer’s comment that the publication of this untargeted metabolomics study in Molecules will have a wide impact in this scientific area. On the other hand, we have already made a reference to a previous metabolic profile study of three different parts of saffron, i.e., tepals, stigmas and stamens by a LC-MS-based methodology (ref. 15).